# Is Thailand Attractive to Japanese Companies?

**Hiroaki Sakurai**

Department of Intercultural Communication, Faculty of Intercultural Studies, Gakushuin Women's College, Tokyo 162-8650, Japan; hiroaki.sakurai@gakushuin.ac.jp; Tel.: +81-3-3203-7199

**Abstract:** This study examines the relationship between the business sentiment of Japanese companies regarding promising or potential countries for investment and macroeconomic statistics, such as economic or population growth in Thailand, using data from the Survey Report on Overseas Business Operations by Japanese Manufacturing Companies from 1992 to 2022. Although investing in Thailand has been popular among Japanese companies since the late 1980s, it has seemingly become relatively inactive in recent years. The present study's results are summarized as follows: First, the business sentiment of Japanese companies has some relationships with relatively short-term economic growth and the business cycle in the short run. Second, business sentiment depends on long-term trends, and this stance may have changed after 2020. Third, other elements, such as minimum wage or fewer young people, do not necessarily have a relationship with business sentiment. Although more studies including capital accumulation or the global value chain should be conducted, improving the sentiments of Japanese businesspersons is desirable.

**Keywords:** investment; Japan; Thailand

## 1. Introduction

Nearly 7000 Japanese companies are located in Thailand, and over 70,000 Japanese citizens reside in Thailand (Ministry of Foreign Affairs of Japan 2024; Japan External Trade Organization 2021). Although Thailand is one of the most popular countries for establishing Japanese-owned factories, in recent years, the business sentiment of Japanese companies toward Thailand has been less favorable than that from a decade ago. An example is the decreasing number of members of the Japanese Chamber of Commerce in Thailand (JCC), established in 1954 by 30 Japanese companies, as shown in Figure 1. The number of JCC members increased to 394 in 1985, when the Plaza Agreement was announced, and 1028 in 1995, soon before the 1997 financial crisis. After the number of members almost flattened during the 2000s, partly because of the effect of the 1997 financial crisis, it increased again during the 2010s. Although the number of JCCs increased until 2019 to 1772 companies, it has begun to decrease. Business sentiment, as surveyed by the JCC, worsened after the coronavirus disease 2019 (COVID-19) outbreak. Other statistics are provided in the Survey Report on Overseas Business Operations by Japanese Manufacturing Companies held by the Japan Bank International Cooperation (hereinafter referred to as the JBIC survey). The question posed by the JBIC is "Please list up to five countries for business development in the medium term (the next three years)," and the ratio of the listed responses is presented. Figure 2 shows the evaluation ratio of Thailand as a promising or potential country. For managers of Japanese companies, Thailand was the second- to fourth-most popular from the middle of the 1990s to the 2010s. However, the latest survey results show that Thailand ranks sixth as a result of a lowered evaluation due to the recent decline.

The summary report of the JBIC survey (Itagaki et al. 2023) summarizes the reasons underlying Thailand's attractiveness from positive and negative perspectives. Positive points for extending business in Thailand are the current market size and supply base for assemblies that typify Thailand's attractiveness. In contrast, the negative points for conducting business in Thailand are that the wages for workers are increased, coupled with the difficulty in hiring

workers due to the labor shortage. Although the summary report of the JBIC survey is correctly gathered from managers' opinions, there is still a gap between managers' sentiments and actual macroeconomic statistics in Thailand, as wage increments and labor shortages do not occur suddenly. If there is a gap between the results of business sentiment and macroeconomic statistics, such as the gross domestic product (GDP) growth rate or depopulation in Thailand, it is useful information for policymakers in Japan and Thailand, because narrowing the gap will entail decreasing additional expenses and increasing mutual profit.

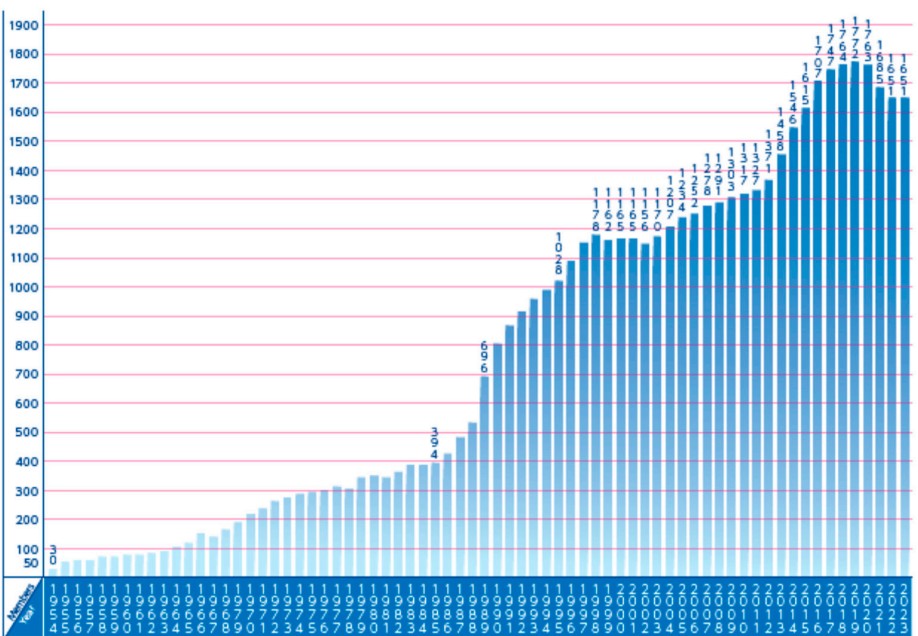

**Figure 1.** Number of members in Japanese Chamber of Commerce (JCC) in Thailand from JCC web page.

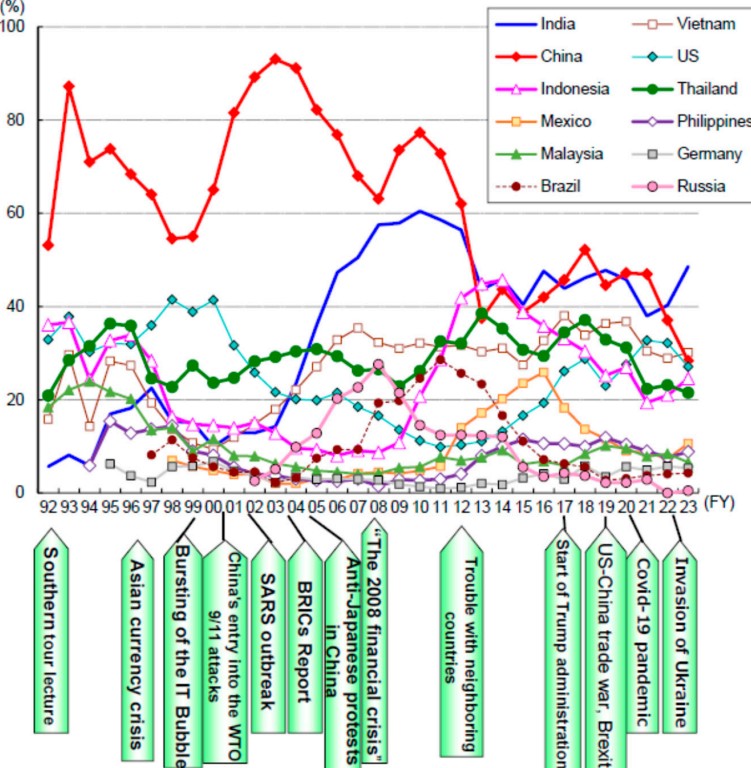

**Figure 2.** Trends in promising/potential countries listed by managers in the JBIC survey from Itagaki et al. (2023).

In fact, the ratio of those answering that Thailand is a promising or potential country differs from its macroeconomic performance. This is obvious by using the HP filter and AR(1) process, as shown in Figure 3. If this statistic (referred to as "attractiveness" herein) is estimated by the previous year, the coefficient is around 0.55. If the AR(1) process is used, the coefficient is approximately 0.6. Hence, the AR(1) process in Figure 3 is depicted as previous year weighted by 60% and present year by 40%. When the HP filter is used, two peaks appear during the 1990s and 2010s, and they both fall sharply after 2018. This comprehensive estimation implies that the business sentiment of Japanese managers may be stable until critical points, rather than changing year by year.

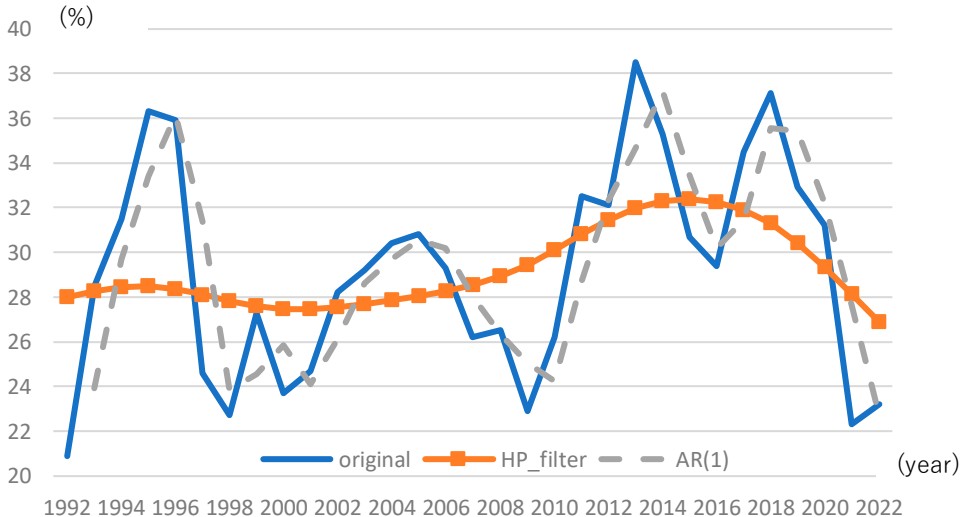

**Figure 3.** Ratio of answering that Thailand is a promising/potential country in the medium term.

However, the reason for outward foreign direct investment by Japanese companies is not necessarily analyzed quantitatively with the economic background of the receiving country. In addition, the relationship with other economic statistics is hardly mentioned, at least in the business, although the business sentiment is often used in the actual business cycle. It is worth introducing the econometric approach by using the business sentiment for better policymaking by the governmental sector and increasing profits for private companies.

This study examines the relationship between the business sentiment of Japanese companies and macroeconomic statistics in Thailand using ordinary least squares (OLS) and vector autoregressive (VAR) models in a time series analysis. By showing the relationship between the business sentiment of Japanese companies and economic statistics in Thailand, policies to make Thailand more attractive will be inferred. The remainder of this paper is organized as follows: Section 2 presents a literature review. Section 3 presents the data, methodology, and estimation results for the poverty ratio. Section 4 presents the data, methodology, and estimation results for social welfare expenditure. Finally, Section 5 concludes the paper.

## 2. Literature Review

Previous studies related to this field have focused on the following three points. First, studies regarding outward foreign direct investment have accumulated from theoretical and practical points of view. Second, the relationship between investment and business sentiment as well as the real business cycle and political uncertainty are studied. Third, country-specific reasons in both Thailand and Japan are researched well, since Thailand experienced a financial crisis in 1997 and Japan experienced the collapse of the bubble economy during the 1990s. The following sections show the previous studies in each field.

### 2.1. Outward Foreign Direct Investment

Reasons for selecting outward foreign direct investment are summarized by Dunning (1981), who stated that ownership, location, and internalization (OLI) influence production decisions. Although the OLI theorem organized important elements well, the extent of each element is obscure. Outward foreign direct investment is divided into two types: horizontal foreign direct investment and vertical foreign direct investment. Horizontal foreign direct investment produces the same products in different countries, mainly to reduce transportation costs or political conflict, and is often seen among developed countries (Markusen 1984; Brainard 1993, 1997). For example, Japanese companies invest in production in the US. In contrast, vertical foreign direct investment divides the process of production. The main reason is the difference in wage, and labor-intensive products are produced in developing countries where workers receive lower wages (Helpman 1984).

The selection of domestic, exported, or overseas investment in production is dependent on productivity (Helpman et al. 2004). Most companies produce and sell within their home countries (e.g., (Bernard et al. 2009) in the US and (Tomiura 2007) in Japan). In addition, the selection between foreign direct investment and outsourcing is analyzed by Antras and Helpman (2004) and Keller and Yeaple (2013). Moreover, the selection of location is also analyzed (Combes et al. 2008).

### 2.2. Business Sentiment and Economic Uncertainty

Business sentiment and investment prospects are related, since investment is decided by board members as representatives in multinational enterprises. Inferring from this fact, business sentiment perspectives are related to investment. In addition, the previous literature has analyzed the relationship between economic uncertainty and investment. Specifically, economic uncertainty is divided into macroeconomic uncertainty and political uncertainty. In this sub-section, previous studies regarding business sentiment, macroeconomic uncertainty, and political uncertainty with investment are summarized as follows.

First, regarding recent studies on business sentiment and consumer behavior, Benhabib and Spiegel (2019) use US consumer data and show that pure optimism boosts real output. Lagerborg et al. (2023) suggest that sentiment-driven impacts are long-lasting. As for the empirical methods, Barsky and Sims (2012) show "animal spirit" shocks and new information shocks by using the VAR model.

Second, as for macroeconomic uncertainty, the relationship between the business cycle and investment is analyzed. The common behavior involves decreasing investment and cutting jobs during a recession (Bernanke 1983; Bloom 2009). Jurado et al. (2015) measured this from the broad perspective of macroeconomic uncertainty. Baker et al. (2016) develop a new index of economic policy uncertainty from newspaper coverage. Ludvigson et al. (2021) analyze future uncertainty and show that the macroeconomic uncertainty in recessions generates endogenous shock, while uncertainty about financial markets is a likely source of output fluctuations.

Third, policy uncertainty is a problem in deciding the investment. This problem happens mainly in emerging markets. The analytical framework uses the imperfect information model (Sims 2003; Mankiew and Reis 2002). Reis (2006) solves the problem of producer-facing costs of acquiring, absorbing, and processing information from a theoretical perspective, showing that producers prefer to set a plan for quantity and that producers only sporadically update the information chosen by agent rationally.

### 2.3. Specific Reasons in Thailand and Japan

Although economic models are described in the previous two sections, country-specific characteristics are also considered in the analysis. The empirical literature on Thailand and Japan is rather interesting, as Thailand experienced the 1997 financial crisis and Japanese managers preferred to hold cash during the collapse of the bubble economy at the start of the 1990s.

Empirical studies on Thailand have mainly focused on uncertainty in the emerging market economy (EME) because Thailand experienced the 1997 financial crisis, and it is still vulnerable to external shocks due to its open economy to the world. According to Apaitan et al. (2022), uncertainty is considered to encompass the following three points. First, behavior is affected by the types of uncertainty shocks, such as global, macroeconomic, financial, economic policy, and financial uncertainty. Jirasakuldech and Emekter (2021) analyzed herding behavior during the crisis in Thailand, and they found that it occurred frequently around the 1997 crisis. The second point entails gaining a deeper understanding of how and when uncertainty shocks are transmitted. Although various mechanisms exist for transmitting uncertainty, channels of investment are important in this study. Examples of such theoretical studies include Bernanke (1983), McDonald and Siegel (1986), and Bloom (2009). A comprehensive analysis of the transmission mechanism will be helpful for policymakers in managing macro and financial conditions in key sectors (Apaitan et al. 2022, p. 936). Third, the crossover effect is demonstrated. Although this is mainly analyzed for the exchange rate and currency crisis, the impact of shocks from abroad is often larger than domestic uncertainty for EMEs, including Thailand (Apaitan et al. 2022).

The attitudes of Japanese companies toward investment are discussed with respect to economic uncertainty. After the 1997 financial crisis, Japanese companies focused on cash holdings rather than borrowing money for investment because of bad loans during the bubble economy from the late 1980s to the start of the 1990s. This subsection refers to Fujitani et al. (2023) to summarize previous studies comprehensively. Sakai (2020) examines Japanese firms during the two decades after the bubble economy collapsed and finds not only that Japanese firms faced financial constraints only during the first decade but also that they neglected investment in the latter decade. Masuda (2015) examines the impact of monetary policy on investment and finds that a contractionary monetary policy tightens corporate liquidity constraints. Ushijima (2020) shows that firms with focused business lines tighten their cash. As for the relationship between the global financial crisis and Japanese companies, Uchino (2013) shows that Japanese companies decreased their investment levels during the 2008 financial crisis, and Tsuruta (2019) shows that adjusting the speed of working capital is slower during the financial crisis. Arbatli Saxegaard et al. (2022) develop the Japanese index, and Fujitani et al. (2023) examine the relationship between the index and investment, finding out that the investment of Japanese companies is affected by economic uncertainty in the US rather than domestic economic uncertainty.

*2.4. Research Gap and Purpose of This Study*

Although many previous studies have been conducted, both theoretically and empirically in each country, the relationship between the business sentiment and the business cycle including the investment is not necessarily obvious. It is important to examine this relationship, since the actual business cycle is measured by the business sentiment. In addition, from an empirical point of view, in Thailand and Japan, no research has been conducted on Japanese business sentiments regarding Thai attractiveness, although many previous studies have been conducted both in Thailand and Japan. This study is worth examining because the results will be useful for not only Japanese managers but also policymakers in both the Japanese and Thai governments. In addition, this study will try to narrow the research gap to show the relationship between business sentiment and investment.

## 3. Methodologies and Data

This section outlines the empirical aspects of this study, including the data for the key variables and relevant methodologies, namely the VAR and OLS models.

*3.1. Data*

In this estimation, five key variables are adopted as endogenous variables. All variables are elements of why Thailand is a promising/potential country according to Itagaki

et al. (2023). The first endogenous variable is the "attractiveness" from the Survey Report on Overseas Business Operations by Japanese Manufacturing Companies by the JBIC (attract). The second is the economic growth rate (growth) from the World Development Indicators of the World Bank (WDI). Economic growth is selected as approximately half of the ratio of responses listing Thailand as a future market in the JBIC survey. The third variable is the increase in the ratio of the working-age population (age), and the fourth is the urban population increase ratio from the WDI, especially as Itagaki et al. (2023) indicate the difficulty of employing labor in Thailand. The fifth is the minimum wage increase ratio from the Thai government, as Itagaki et al. (2023) depict the wage hike as the difficulty in investing in Thailand. Data descriptions of the five variables are presented in Table 1.

**Table 1.** Data description.

|  | **Attract** | **Growth** | **Age** | **Urban** | **Wage** |
|---|---|---|---|---|---|
| obs. | 31 | 31 | 31 | 31 | 31 |
| mean | 0.292 | 0.035 | 0.009 | 0.027 | 0.044 |
| min | 0.209 | −0.086 | −0.005 | 0.015 | −0.018 |
| max | 0.385 | 0.100 | 0.024 | 0.047 | 0.395 |

Source: Survey Report on Overseas Business Operations by Japanese Manufacturing Companies (JBIC), World Development Indicators (World Bank), and Thai government.

### 3.2. Methodologies

We conduct estimations using the VAR (OLS) model to depict the short (long)-term relationship. The VAR model is suitable for determining the relationships among the variables of interest and facilitates the tracing of the dynamic responses of the variables to an exogenous shock. Conversely, the OLS model is used for determining the stationary relationship during the entire period.

Before constructing the VAR and OLS models, we conduct unit root tests for stationarity. Augmented Dickey–Fuller (ADF) and Phillips–Perron (PP) tests are used to assess whether these statistics have unit roots. The test for stationarity involves the null hypotheses of unit roots on the values and their first difference, including both "intercept" and "trend and intercept".

The VAR model is constructed as per the following equation:

$$
\begin{bmatrix} D(attract)_t \\ D(growth)_t \\ D(age)_t \\ D(urban)_t \\ D(wage)_t \end{bmatrix} = \begin{bmatrix} \alpha_{10} \\ \alpha_{20} \\ \alpha_{30} \\ \alpha_{40} \\ \alpha_{50} \end{bmatrix} + \begin{bmatrix} \beta_{11} & \beta_{12} & \beta_{13} & \beta_{14} & \beta_{15} \\ \beta_{21} & \beta_{22} & \beta_{23} & \beta_{24} & \beta_{25} \\ \beta_{31} & \beta_{32} & \beta_{33} & \beta_{34} & \beta_{35} \\ \beta_{41} & \beta_{42} & \beta_{43} & \beta_{44} & \beta_{45} \\ \beta_{51} & \beta_{52} & \beta_{53} & \beta_{54} & \beta_{55} \end{bmatrix} \begin{bmatrix} D(attract)_{t-1} \\ D(growth)_{t-1} \\ D(age)_{t-1} \\ D(urban)_{t-1} \\ D(wage)_{t-1} \end{bmatrix} + \begin{bmatrix} \varepsilon_{1t} \\ \varepsilon_{2t} \\ \varepsilon_{3t} \\ \varepsilon_{4t} \\ \varepsilon_{5t} \end{bmatrix} \quad (1)
$$

where $D(\text{--})_t$ denotes the first difference in period $t$. The other terms in the equation are defined as flows: $\alpha$ denotes the constant term, $\beta$ represents endogenous variables, and $\varepsilon$ indicates the error term.

The OLS model is defined as follows:

$$
D(attract)_t = \beta_0 + \beta_1 D(growth)_t + \beta_2 D(age)_t + \beta_3 D(urban)_t + \beta_4 D(wage)_t + \varepsilon_t \quad (2)
$$

If cointegrating relations are found, then the cointegrated VAR model is also used.

## 4. Estimation Results and Discussion

This section presents the results of the estimation, which are divided into three parts. First, the results of the unit root tests are provided. Second, the estimation results of the VAR model and Granger causality tests are shown. Third, the OLS estimation results are depicted as long-term relationships. Although an attempt is made to examine the

cointegrated VAR model, no cointegrating relations are found in this model. Therefore, only the VAR and OLS models are presented herein.

### 4.1. Unit Root Tests

The results of the unit root tests are presented in Table 2. Only economic growth (growth) and minimum wage (wage) are white noise, I(0), and the other three variables, namely, attractiveness (attract), age from 15 to 64 (age), and urban population increase ratio (urban), have unit roots I(1) or I(2). When constructing the VAR model with variables with unit roots, the first difference is used. In addition, the residual OLS term must be I(0), with white noise in using the OLS model and with variables having a unit root.

**Table 2.** Estimation results of unit root tests.

| | ADF | | PP | |
|---|---|---|---|---|
| **attract: I(1)** | | | | |
| | intercept | intercept and trend | intercept | intercept and trend |
| level | −2.953 * | −2.674 | −3.142 ** | −2.911 |
| first difference | −4.952 *** | −4.856 *** | −5.606 *** | −5.341 *** |
| **growth: I(0)** | | | | |
| | intercept | intercept and trend | intercept | intercept and trend |
| level | −3.535 ** | −3.723 ** | −3.469 ** | −3.688 ** |
| first difference | −3.078 ** | −5.682 *** | −7.228 *** | −7.093 *** |
| **age: I(2)** | | | | |
| | intercept | intercept and trend | intercept | intercept and trend |
| level | 1.010 | −2.439 | −0.474 | −1.897 |
| first difference | −3.046 ** | −3.261 * | −2.286 | −2.142 |
| second difference | | −6.603 *** | −5.150 *** | −5.533 *** |
| **urban: I(1)** | | | | |
| | intercept | intercept and trend | intercept | intercept and trend |
| level | −1.073 | −1.244 | −1.317 | −1.244 |
| first difference | −4.026 *** | −4.131 ** | −4.040 *** | −4.067 ** |
| **wage: I(0)** | | | | |
| | intercept | intercept and trend | intercept | intercept and trend |
| level | −5.372 *** | −5.258 *** | −5.497 *** | −5.310 *** |
| first difference | - | - | - | - |

Note: ***, **, and * indicate significance at the 1%, 5%, and 10% levels, respectively. Source: Author's calculations.

### 4.2. VAR Model and Granger Causality Test

The results of the VAR model shown in Equation (1) are reported in Table 3, and the pairwise Granger causality tests are provided in Table 4. D(--) indicates the first difference, and (−1) means the previous period.

**Table 3.** Estimation results of the VAR model.

|  | D(attract) | D(age) | D(growth) | D(urban) | D(wage) |
|---|---|---|---|---|---|
| D(attract(−1)) | 0.181 | −0.004 | 0.192 | −0.017 | 0.292 |
|  | (0.184) | (0.002) * | (0.240) | (0.025) | (0.406) |
| D(age(−1)) | 12.290 | 0.526 | −1.651 | −0.082 | 55.861 |
|  | (14.374) | (0.173) *** | (18.718) | (1.915) | (31.691) * |
| D(growth(−1)) | 0.259 | 0.001 | −0.365 | 0.007 | −0.297 |
|  | (0.155) | (0.002) | (0.202) * | (0.021) | (0.341) |
| D(urban(−1)) | 1.118 | −0.010 | 0.829 | 0.225 | −5.551 |
|  | (1.518) | (0.018) | (1.977) | (0.202) | (3.347) |
| D(wage(−1)) | 0.082 | −0.001 | 0.073 | −0.002 | −0.497 |
|  | (0.076) | (0.001) | (0.100) | (0.010) | (0.168) *** |
| C | 0.011 | 0.000 | −0.006 | 0.000 | 0.048 |
|  | (0.016) | (0.000) ** | (0.020) | (0.002) | (0.035) |
| Adj. R−squared | 0.063 | 0.288 | −0.039 | −0.111 | 0.287 |

Note: D(--) indicates the first difference, and (−1) indicates the previous period. Standard errors are shown in parentheses. ***, **, and * indicate significance at the 1%, 5%, and 10% levels, respectively. Source: Author's calculation.

**Table 4.** Estimation results of the Granger causality tests.

| Null Hypothesis: | Observations | F−Statistic |
|---|---|---|
| D(AGE) does not Granger Cause D(ATTRACT) | 29 | 0.640 |
| D(ATTRACT) does not Granger Cause D(AGE) | 29 | 2.661 |
| D(GROWTH) does not Granger Cause D(ATTRACT) | 29 | 3.717 * |
| D(ATTRACT) does not Granger Cause D(GROWTH) | 29 | 0.594 |
| D(URBAN) does not Granger Cause D(ATTRACT) | 29 | 0.700 |
| D(ATTRACT) does not Granger Cause D(URBAN) | 29 | 0.512 |
| D(WAGE) does not Granger Cause D(ATTRACT) | 29 | 2.433 |
| D(ATTRACT) does not Granger Cause D(WAGE) | 29 | 0.088 |
| D(GROWTH) does not Granger Cause D(AGE) | 29 | 0.257 |
| D(AGE) does not Granger Cause D(GROWTH) | 29 | 0.027 |
| D(URBAN) does not Granger Cause D(AGE) | 29 | 0.082 |
| D(AGE) does not Granger Cause D(URBAN) | 29 | 0.008 |
| D(WAGE) does not Granger Cause D(AGE) | 29 | 0.176 |
| D(AGE) does not Granger Cause D(WAGE) | 29 | 2.748 |
| D(URBAN) does not Granger Cause D(GROWTH) | 29 | 0.083 |
| D(GROWTH) does not Granger Cause D(URBAN) | 29 | 0.098 |
| D(WAGE) does not Granger Cause D(GROWTH) | 29 | 0.408 |
| D(GROWTH) does not Granger Cause D(WAGE) | 29 | 1.718 |
| D(WAGE) does not Granger Cause D(URBAN) | 29 | 0.007 |
| D(URBAN) does not Granger Cause D(WAGE) | 29 | 2.809 |

Note: D(--) indicates the first difference. Standard errors are shown in parentheses. * indicates significance at the 10% levels. Source: Author's calculations.

The VAR model results indicate the following two points. First, almost all relationships are estimated to be insignificant. There are five exceptions: previous attractiveness to the present age, previous age to present age, previous age to present wage, previous growth to present growth, and previous wage to present wage. As most variables in social science have a relationship with those in the previous period, it is natural that the previous period estimates the present period effectively. Second, no variables are effectively estimated with the attractiveness in the present time. In contrast, the results of the Granger causality tests in Table 4 show that only economic growth to attractiveness is effectively estimated.

As the results of the VAR model and Granger causality test differ in the relationship between economic growth (business cycle) and attractiveness (business sentiment), we construct the VAR model and undertake the Granger causality test with a focus on these two variables. As both variables are I(0), the estimation can be held by the level series. The estimation results of the VAR model are shown in Table 5, and the pairwise Granger causality tests are provided in Table 6.

**Table 5.** Estimated results of the VAR model between attractiveness and growth.

|  | **ATTRACT** | **GROWTH** |
|---|---|---|
| ATTRACT(−1) | 0.467 | −0.063 |
|  | (0.139) *** | (0.176) |
| GROWTH(−1) | 0.404 | 0.401 |
|  | (0.143) *** | (0.180) ** |
| C | 0.143 | 0.037 |
|  | (0.04) *** | (0.051) |
| Adj. R-squared | 0.439 | 0.093 |

Note: (−1) indicates the previous period. Standard errors are shown in parentheses. ***and ** indicate significance at the 1%and 5% levels, respectively. Source: Author's calculations.

**Table 6.** Estimated results of the Granger causality tests between attractiveness and growth.

| **Null Hypothesis:** | **Observations** | **F-Statistic** |
|---|---|---|
| GROWTH does not Granger Cause ATTRACT | 30 | 8.051 *** |
| ATTRACT does not Granger Cause GROWTH | 30 | 0.128 |

Note: *** indicates significance at the 1% level. Source: Author's calculations.

The VAR model in Table 5 indicates that positive effectiveness estimates the relationship between economic growth in the previous period and attractiveness in the present period. In addition, the results of the Granger causality tests effectively estimate the relationship between economic growth in the previous period and attractiveness in the present period.

As the estimation result is clear in this relationship, the impulse response is additionally examined. The result is shown in Figure 4, depicting that the effect is bigger and more stable until the third year, which is the same result as observed for the survey question "potential for the next three years". In this regard, business sentiment appears to predict the business cycle correctly in the short term.

From the estimation results in this subsection, it is inferred that the attractiveness of Thailand as a potential country in the relatively short term will have a relationship with economic growth or business cycles, and that it will have almost no relationship with other elements, such as minimum wage or depopulation.

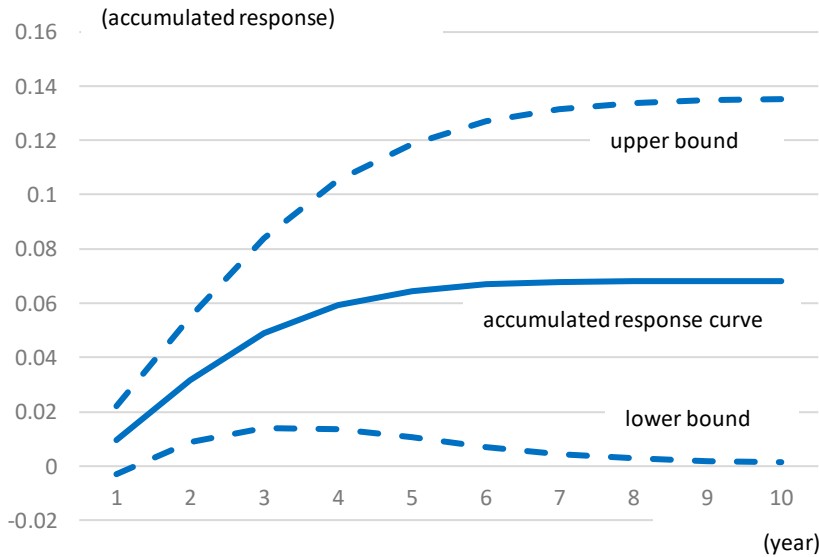

**Figure 4.** Accumulated impulse response from growth to attractiveness.

*4.3. OLS Estimation*

Finally, OLS estimations are examined to determine the relationship between attractiveness and other variables. Although we tried to examine the OLS model with attractiveness as the dependent variable, attractiveness was not significantly estimated by the other variables. Instead, we define two dummy variables. One dummy variable is whether the data are from after 2009 and the global financial crisis. The other dummy variable is whether the data are from after 2020, during the COVID-19 crisis, along with the two peaks evident in Figure 3. Another estimation uses constant terms and AR(1). The two estimated equations are described in Table 7.

**Table 7.** Estimation results.

| | Dependent Variable: ATTRACT Estimation Period: 1992–2022 | |
| --- | --- | --- |
| | ① | ② |
| DUMMY20 | −0.064 | |
| | (0.029) ** | |
| DUMMY09 | 0.040 | |
| | (0.017) ** | |
| AR(1) | | 0.601 |
| | | (0.163) *** |
| C | 0.280 | 0.286 |
| | (0.011) *** | (0.016) *** |
| Adjusted R-squared | 0.159 | 0.280 |
| Durbin-Watson stat | 1.134 | 1.588 |
| Residual unit root test | I(0) | I(0) |

Notes: Standard errors are shown in parentheses. *** and ** indicate significance at the 1% and 5% levels, respectively. Source: Author's calculations.

Estimated Equation (1) shows that the attractiveness of Thailand changed roughly three times. Until 2008, it was 0.280, indicating that approximately 28% of the managers of Japanese companies evaluated Thailand as a potential country. Attractiveness increased to 0.320 (=0.280 + 0.040) after the 2009 crisis, meaning that more managers of Japanese companies found Thailand attractive during the 2010s. It decreased to 0.256 (=0.280 + 0.040 − 0.064) after 2020. Estimated Equation (2) shows that the attractiveness sentiment is affected by the previous year, which is explainable by the fact that we decide on our activities from our knowledge and experience.

From the OLS estimations, it is inferred that managers' minds will change during a major crisis in a decade, and after making a decision, it will continue for a while. This will not always change soon because of the economic situation.

## 5. Conclusions

This study examines the relationship between the business sentiment of Japanese companies regarding Thailand as a promising/potential country and macroeconomic statistics including GDP growth, population change rate in the age between 15 and 64, urban population change, and minimum wage using the JBIC survey from 1992 to 2022. Although investing in Thailand has been popular among Japanese companies since the late 1980s, investment has been relatively inactive in recent years, regardless of Thailand's macroeconomic and financial stability. It is important for both businesspersons and policymakers in Thailand and Japan to find out why the expectation for expanding business in Thailand has decreased, because Japanese companies already have a huge base of manufacturing production systems in Thailand.

The results are summarized as follows. First, the business sentiment of Japanese companies has some relationships with relatively short-term economic growth or business cycles in the short run. Second, business sentiment depends on long-term trends in the long run, and this stance may have changed after 2020. Third, other elements, such as minimum wage or fewer young people, do not necessarily have a relationship with business sentiment. These results are consistent with previous studies in Thailand on herding behavior and in Japan on preferences for holding cash.

The policy implications of the results are relevant to businesspersons in Japanese firms, as well as policymakers in Thailand and Japan. First, businesspersons in Japanese firms should acquire more confidence in Thailand by visiting and meeting with policymakers and businesspersons in Thailand. Second, for policymakers in Thailand, the focus should be on spreading information on the economic stability of and policy efforts in Thailand. Third, policymakers in Japan can guide the promotion of interaction and mutual understanding between Thailand and Japan.

As this study uses limited statistics, more research that includes elements of the global value chain or capital accumulation is necessary to understand the reasons for the lowering of the business sentiment in Japanese companies toward Thailand. More research in this field will facilitate mutual understanding between Thailand and Japan, which will improve the sentiment of Japanese businesspersons and further allow for mutual profit.

**Funding:** This work was supported by JSPS KAKENHI Grant Number JP24K16375.

**Institutional Review Board Statement:** Not applicable.

**Informed Consent Statement:** Not applicable.

**Data Availability Statement:** Data for the study was gathered from World Development Indicators, Royal Thai Government, and Japan Bank for International Cooperation.

**Conflicts of Interest:** The author declares no conflict of interest.

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
