# Peer review of "Is Thailand Attractive to Japanese Companies?"

_economies, doi:10.3390/economies12050122_

Round 1

Reviewer 1 Report

Comments and Suggestions for Authors

This study examines the relationship between the business sentiment of Japanese companies of Thailand as a promising/potential country and macroeconomic statistics including GDP growth, population change rate in the age between 15 and 64, urban population change, and minimum wage using the JBIC survey from 1992 to 2022.

The purpose is very interesting and actual. What is harmful is the weak base of scientific approaches.

The paper is processed clearly, the individual parts follow each other and complement each other appropriately. The author uses current sources of knowledge.

I have some recommendations for improvement.

Literature review:

Enlarge this section. Add some other authors and their view on the analysed area. Develop this section in more depth with the additional expert opinions.

Add other sources. The current number of sources is insufficient.

Author Response

Thank you for your good and warm comments and suggestions.

  1. Since there is no mention about the research in the introduction, I added on the importance of contributing research.

  1. Since I received your comment, I knew that literature review is insufficient.
  • I added on the previous literature of the outward foreign direct investment as the section 2.1.
  • I also increase the number of the previous literature of the economic uncertainty to make up for the scientific approach in the section 2.2.
  • Country specific researches are summarized in the section 2.3 and show the research gap in the section 2.4.

  1. I added on the sentence about research results are suitable for previous studies and further research will contribute the development of the scientific methodology by using business sentiment.

I will refine again if there are still some problems.

I would appreciate if you could indicate.

Thanks to your suggestions, this manuscript has been much better.

Thank you for your taking time.

Sincerely,

Reviewer 2 Report

Comments and Suggestions for Authors

Dear/ Author/s,

The objective may be interesting from a local point of view, but I believe that a theoretical framework should be created, to point out how to make countries attractive so that others with more resources will want to invest in them. It would be necessary to point out a clear research gap and justify the interest of studying this specific case, framed within a broader scientific problem. With this, the authors would have the introduction. Subsequently, a bibliographic review of the variables to be studied in the research problem would be carried out. With all the above, the conclusions are practically a summary of the results obtained, without any contrast with the theoretical framework, so that the work does not present any type of theoretical implications, which could suppose any scientific contribution.

Best Regards

Author Response

(The authors gave the same response as above.)

Round 2

Reviewer 2 Report

Comments and Suggestions for Authors

Dear Sirs,

The article has improved somewhat with respect to its first version. The research gap is noted at the end of the literature review, an aspect that should appear at the end of the introduction. Given this situation, a single introductory section could be created. The conclusions remain practically in the same state as the previous version, without any reference to contrast the results with the theoretical framework.

Best Regards.

Author Response

Thank you for your good comment.
I add on a single introductory section to appear an aspect to make the country more attractive.
I also deleted the conclusion without any reference to contrast the results with the theoretical framework.
Thanks to your comment, I could learn how to construct the introduction section.
I will refine again if I have some misunderstanding.

Sincerely,